# Synergistic Effect of Human Papillomavirus and Environmental Factors on Skin Squamous Cell Carcinoma, Basal Cell Carcinoma, and Melanoma: Insights from a Taiwanese Cohort

**DOI:** 10.3390/cancers16122284

**Published:** 2024-06-20

**Authors:** Chun-Chia Chen, Ci-Wen Luo, Stella Chin-Shaw Tsai, Jing-Yang Huang, Shun-Fa Yang, Frank Cheu-Feng Lin

**Affiliations:** 1Institute of Medicine, Chung Shan Medical University, Taichung 40201, Taiwan; chenjica@gmail.com (C.-C.C.); cshe961@csh.org.tw (J.-Y.H.); ysf@csmu.edu.tw (S.-F.Y.); 2Division of Plastic Surgery, Department of Surgery, Chi Mei Medical Center, Tainan 71004, Taiwan; 3Department of Medical Research, Tungs’ Taichung MetroHarbor Hospital, Taichung 43503, Taiwan; t14825@ms3.sltung.com.tw; 4Superintendent Office, Tungs’ Taichung MetroHarbor Hospital, Taichung 43503, Taiwan; 5Department of Post-Baccalaureate Medicine, College of Medicine, National Chung Shin University, Taichung 402202, Taiwan; 6Department of Medical Research, Chung Shan Medical University Hospital, Taichung 40201, Taiwan; 7School of Medicine, Chung Shan Medical University, Taichung 40201, Taiwan; 8Department of Surgery, Chung Shan Medical University Hospital, Taichung 40201, Taiwan

**Keywords:** human papillomavirus (HPV), skin cancer, squamous cell carcinoma (SCC), basal cell carcinoma, melanoma, National Health Insurance Research Database (NHIRD)

## Abstract

**Simple Summary:**

The study examined the relationship between Human Papillomavirus (HPV) and various types of skin cancer, including squamous cell carcinoma, basal cell carcinoma, and melanoma. It aimed to assess how HPV influenced the likelihood of developing these cancers, considering factors such as age, gender, urbanization, and existing health conditions. Through statistical models, the research quantified the risk of skin cancer in individuals with HPV compared to those without, underscoring how behaviors related to sun exposure and protection practices could alter these risks. The findings were crucial for guiding prevention and treatment strategies and were of significant interest to medical professionals and public health policymakers. Overall, the study enhanced our understanding of the risks associated with HPV, potentially leading to more effective health interventions.

**Abstract:**

Human papillomavirus (HPV) has been implicated in various cancers, including those affecting the skin. The study assessed the long-term risk of skin cancer associated with HPV infection in Taiwan region, using data from the National Health Insurance Research Database between 2007 and 2015. Our analysis revealed a significant increase in skin cancer risk among those with HPV, particularly for squamous cell carcinoma (SCC), the subtype with the highest observed adjusted hazard ratio (aHR) = 5.97, 95% CI: 4.96–7.19). The overall aHR for HPV-related skin cancer was 5.22 (95% CI: 4.70–5.80), indicating a notably higher risk in the HPV-positive group. The risk of skin cancer was further stratified by type, with basal cell carcinoma (aHR = 4.88, 95% CI: 4.14–5.74), and melanoma (aHR = 4.36, 95% CI: 2.76–6.89) also showing significant associations with HPV. The study also highlighted regional variations, with increased risks in southern Taiwan and the Kaohsiung-Pingtung area. Key findings emphasize the importance of sun protection, particularly in regions of high UV exposure and among individuals in high-risk occupations. This research contributes to a better understanding of the complex interactions between HPV and skin cancer risk, reinforcing the importance of preventive strategies in public health.

## 1. Introduction

Human papillomavirus (HPV) is well recognized for its role in causing certain types of cancers, particularly cervical and oropharyngeal cancers [1]. While HPV is primarily linked to these mucosal cancers, its association with skin cancers is less direct but increasingly studied [2,3]. HPV infections are generally asymptomatic, with warts being the most common clinical manifestation on the skin [4,5]. Although 90% of HPV infections will clear up on their own within two years, the risk of developing cancer increases with prolonged latency and frequent infections [5]. Transmission commonly occurs through sexual activity, with about 75% of sexually active individuals experiencing HPV infection at some point in their lives [6]. According to the World Health Organization (WHO), approximately 13 million people are infected with HPV in the United States each year, with 36,000 cases leading to cancer [7]. Globally, one-third of males are infected with HPV, while around 20% of females are affected by HPV-related cervical cancer, resulting in 625,600 cases of HPV-related cancers worldwide [7].

Skin cancer poses a global threat to public health, affecting millions of individuals worldwide and imposing a significant economic burden on healthcare systems [8]. HPV has been extensively studied and confirmed to increase the risk of skin cancer [9]. Skin cancer is primarily classified into three major types: squamous cell carcinoma (SCC), basal cell carcinoma (BCC), and melanoma [10]. Caucasians have a higher risk of developing skin cancer compared to Asians and Africans [11]. Skin cancer is the most common malignancy globally, with its incidence showing a rising trend over the years. It is estimated that there will be 76,380 new cases of melanoma in 2016 [12]. UV radiation is a known risk factor for skin cancer [13], but research has also demonstrated a correlation between UV and HPV [2], leading to an increased risk of skin cancer. Certain types of β-HPV activate the transforming properties of E6 and E7 [14], which, respectively, bind to the products of tumor suppressor genes p53 and Rb1 [15], resulting in malignant tumor formation. UVB-induced mutation of p53 is considered a major risk factor [16].

A retrospective study conducted in Greece evaluated melanoma biopsy specimens and detected high-risk HPV types [17]. An epidemiological study in Australia suggested that HPV may contribute to the formation of SCC [18]. In European populations, research indicates that the presence of follicular melanocytes in active specimens may be attributed to HPV infection [19]. In a region-wide cohort study conducted in Taiwan, patients with HPV infection had a 2.45-fold increased risk of developing skin cancer compared to matched controls [9]. Based on the above, our research aims to assess the long-term risk of skin cancer among the HPV-infected population in Taiwan, as well as the correlation between HPV infection and the risk of various types of skin cancer.

## 2. Materials and Methods

### 2.1. Database Source

The National Health Insurance Research Database (NHIRD) in Taiwan region is an exemplary source of population-level data. The Health and Welfare Data Center (HWDC) was been established by the Ministry of Health and Welfare (MOHW) in Taiwan area to serve as a centralized data storage site for managing and analyzing NHIRD and approximately 70 other health-related databases [20]. The Taiwan Cancer Registry (TCR), administered by the Ministry of Health and Welfare in Taiwan, is a comprehensive regional population registry that supplies essential data on cancer incidence, management, and survival rates in Taiwan [21]. To enhance data privacy protection, researchers need to establish a remote connection to MOHW servers for on-site analysis at the HWDC. The National Health Insurance (NHI) program was established in 1995 as a single-payer health insurance system by the Taiwanese government. Its primary objective is to improve the accessibility and affordability of healthcare services. Under the NHI, the Taiwanese government has entered into contractual agreements with the majority of healthcare facilities in the country. Physicians are required to upload claims data for each patient visit to the National Health Insurance Ministry. Notably, Taiwan’s healthcare system allows patients with non-emergency health concerns to seek specialist care without the need for referrals from general practitioners. This flexibility enables patients to choose between local private or public clinics and directly access specialists at hospital outpatient departments. As of 2017, approximately 93% of healthcare facilities in Taiwan were contracted with the NHI, with only a few self-pay private clinics being exceptions. The NHI provides universal healthcare coverage, encompassing all necessary medical expenses, including outpatient visits, inpatient care, prescriptions, traditional Chinese medicine treatments, dental services, surgeries, and diagnostic investigations such as X-rays and magnetic resonance imaging. Since its establishment, the NHI has achieved a consistently high coverage rate, starting at 92% and ultimately extending to cover 99.9% of the Taiwanese population by the end of 2014.

### 2.2. Study Population

This study obtained data on the characteristics of HPV patients from the medical records of 26 million Taiwanese individuals in the NHIRD database between 2007 and 2015. HPV diagnosis was defined by ICD-9-CM codes: 078.10 (viral warts, unspecified), 078.19 (other specified viral warts), 078.1× (various forms of viral warts caused by HPV), 078.11 (condyloma acuminatum/genital warts), and 079.4× (unspecified viral and chlamydial infections) [22,23]. A total of 1,103,771 individuals with HPV and 25,462,267 individuals without HPV were identified (Figure 1). We excluded individuals with a history of HPV prior to 2007 and individuals who had cancer before the diagnosis of HPV. The exclusion of individuals with a recorded diagnosis of HPV prior to 2007 was strategic, allowing us to clearly define the cohort’s exposure period starting from 2007, thereby establishing a baseline from which to monitor new HPV diagnoses and subsequent skin cancer development. By setting 2007 as the exposure inception point, we aimed to differentiate between those exposed to HPV from those unexposed within the defined timeframe, ensuring a clear temporal relationship between HPV exposure and the development of skin cancer outcomes. Similarly, the exclusion of individuals with any cancer diagnosis prior to their HPV diagnosis was critical to eliminate confounding effects that pre-existing cancer might have on the study outcomes. This approach ensures that the observed associations between HPV and skin cancer are not influenced by previous cancers, which could skew the results and compromise the study’s objectives to assess the direct impact of HPV on skin cancer risk. After implementing a 1:2 age and gender matching, where each HPV-positive individual (exposed group) was compared to two controls without a recorded HPV diagnosis (unexposed group) from the same dataset, we included 939,874 cases in the HPV group and 1,879,748 controls in the non-HPV group.

Subsequently, the definition of skin cancer was based on the diagnosis of skin cancer after the index date, using ICD-10-CM codes: C43 and C44. Identification and coding of the primary site (topography) and histology (morphology) of malignant tumors were based on the International Classification of Diseases for Oncology, Third Edition (ICD-O-3) published by the World Health Organization in 2000. Cases with malignant behavior codes were included in TCR. We classified them as follows: carcinoma: 8010–8576; squamous cell carcinoma: 8050–8078, 8083–8084; basal cell carcinoma: 8090–8098; melanoma: 8720–8790.

### 2.3. Accounting for Occupational Exposure in Matching Criteria

Recognizing the significance of occupational sun exposure as a potential confounding factor in the relationship between HPV and skin cancer, our study meticulously accounted for the participants’ jobs by matching based on the type of insurance coverage. In Taiwan, insurance coverage is closely tied to one’s occupation, with specific insurance categories for Labor, Agricultural, and Fishery workers, among others. This categorization allows for control of occupational exposure, particularly to sunlight, which is more prevalent in certain job sectors. By utilizing this matching criterion, we ensured that the comparison between the HPV-positive group and the control group accounted for differences in occupational sun exposure risks. This approach enhances the reliability of our findings by ensuring that any observed differences in skin cancer incidence are more likely attributable to HPV status rather than confounding by occupational exposure.

### 2.4. Comorbidities

This study employed the subsequent comorbidities [24,25] as covariates for regression adjustment in order to mitigate potential confounding effects: Ischemic heart disease (ICD9: 414.01–414.9), Hypertension (ICD9: 401.1–405.99), Stroke (ICD9: 430–438), Diabetes mellitus (ICD9: 250.0–250.9), Abnormal liver function (ICD9: 790.4), Renal failure (ICD9: 584.9), GI bleeding (ICD9: 578.9), Hyperlipidemia (ICD9: 272.0–272.4), Chronic kidney diseases (ICD9: 585.1–585.9), COPD (Chronic Obstructive Pulmonary Disease) (ICD9: 491.21–491.22, 492.8, 496), Peptic ulcer (ICD9: 531.00–534.91), Gout (ICD9: 274.00–274.9).

### 2.5. Statistical Analysis

To enhance comparability and reliability, we accounted for known confounding factors and adjusted for multiple variables to improve the accuracy of the results. Categorical variables like sex, income, and urbanization level were compared between HPV and Non-HPV patients using the chi-square test. The Shapiro–Wilk test was employed to assess abnormal distribution (*p* < 0.05). Differences in continuous variables between the groups were evaluated using the Wilcoxon rank-sum test. In the context of case–control sampling, we employed multivariate stratified Cox regression models to calculate the hazard ratio (HR) and 95% confidence interval (CI) for secondary outcomes. We presented a simple set of estimation equations for inference and suggested a potentially more efficient approach. The framework also justified the inclusion of status as a covariate in regression models. We used a multivariate Cox regression model to assess the risk faced by each participant. The results were reported in terms of hazard ratio (HR) and 95% CI. Data analysis was performed using SAS 9.4 software, and statistical significance was defined as *p* < 0.05.

## 3. Results

### 3.1. Baseline Characteristics

Table 1 presents the demographic characteristics of the study population and the inter-group differences among participants with and without HPV. Due to matching, there were no statistically significant differences in gender and age. Regarding the degree of urbanization, it was observed that a majority of HPV patients resided in the highest urbanization level areas (319,694 individuals, 34.01%). In contrast, most non-HPV patients resided in the second-highest urbanization level areas (587,017 individuals, 31.23%), with a statistically significant difference. In terms of residential area, both groups had the highest number of individuals from the Taipei area. Still, HPV patients (395,713 individuals, 42.10%) had a higher proportion compared to non-HPV patients (671,431 individuals, 35.72%), with a statistically significant difference. Concerning insurance coverage, HPV patients were predominantly covered by Labor insurance (65.84%), followed by Unemployment (13.00%), while non-HPV patients were also mainly covered by Labor insurance (62.94%), followed by Unemployment (15.66%), showing a statistically significant difference. Among comorbidities, there were no statistically significant differences in Stroke and Abnormal liver function. Additionally, non-HPV patients (4.98%) had a higher proportion of Diabetes mellitus compared to HPV patients (5.14%), and HPV patients had higher proportions in other diseases, reaching statistically significant differences.

### 3.2. Risk Relationship between HPV and Cumulative Incidence of Skin Cancer

The association between HPV and the cumulative incidence of skin cancer was examined using the Kaplan–Meier method. The data revealed that individuals with HPV exhibited a significantly elevated risk of developing skin cancer over time compared to those without HPV. The study found that the overall hazard ratio for skin cancer associated with HPV was 4.88 (*p* < 0.0001), indicating a significantly higher risk among the HPV-positive group (Figure 2). The risk was further analyzed by skin cancer subtypes, revealing significant associations with squamous cell carcinoma (Figure 3; HR = 5.27, *p* < 0.0001), basal cell carcinoma (Figure 4; HR = 4.74, (*p* < 0.0001), and melanoma (Figure 5; HR = 4.09, *p* < 0.0001).

### 3.3. Cox Regression

Table 2 presented the risk assessment of HPV on skin cancer after adjusting for sex, age, urbanization, area, insurance coverage, and co-morbidity. The results indicated that the hazard ratio (HR) for HPV-associated skin cancer was 5.22 (95% CI: 4.70–5.80), which was statistically significant, suggesting a higher risk of skin cancer among patients with HPV. Regarding gender, males exhibited a higher risk of developing skin cancer compared to females, with an HR of 1.20 (95% CI: 1.10–1.34). In terms of age, compared to the 20–40 years age group, the HR for the age group under 20 years was 0.14 (95% CI: 0.08–0.25), for the 40–60 years age group was 4.15 (95% CI: 3.36–5.15), for the 60–80 years age group was 19.55 (95% CI: 15.84–24.13), and for the age group over 80 years was 58.20 (95% CI: 46.21–73.30), all of which were statistically significant, indicating an increased risk of skin cancer with advancing age. There was no statistically significant difference observed based on the urbanization level. Regarding area, compared to Taipei, Southern Taiwan exhibited an HR of 1.85 (95% CI: 1.57–2.18), Kaohsiung-Pingtung has an HR of 1.74 (95% CI: 1.49–2.02), and Eastern Taiwan has an HR of 1.58 (95% CI: 1.14–2.18), all showing significant differences, suggesting a higher risk of skin cancer in the southern and eastern regions of Taiwan. In terms of insurance coverage, compared to Labor insurance, the Agricultural, Irrigation, and Fishery Associations group has an HR of 1.68 (95% CI: 1.45–1.95), which was statistically significant, indicating a higher risk of skin cancer among individuals engaged in agricultural, irrigation, and fishery. Regarding co-morbidities, hypertension exhibited an HR of 1.26 (95% CI: 1.12–1.42), which was statistically significant, suggesting a higher risk of skin cancer among participants with hypertension in this study.

Table 3 further analyzed the relationship between HPV and different types of skin cancer through Cox regression, adjusting for sex, age, urbanization, area, insurance coverage, and co-morbidities. The results indicated that the adjusted HR for HPV-associated carcinoma was 5.40 (95% CI: 4.82–6.06), for squamous cell carcinoma was 5.97 (95% CI: 4.96–7.19), for basal cell carcinoma was 4.88 (95% CI: 4.14–5.74), for other carcinoma was 5.96 (95% CI: 4.32–8.22), and for melanoma was 4.36 (95% CI: 2.76–6.89); all of which were statistically significant. Among these, the adjusted HR for HPV-associated squamous cell carcinoma was the highest.

## 4. Discussion

This study rigorously analyzed the impact of HPV on the risk of developing various types of skin cancer using multiple Cox regression models, adjusting for critical factors such as sex, age, urbanization, area, insurance coverage, and co-morbidity. The results consistently demonstrated a significantly elevated HR for skin cancer in individuals with HPV across all analyzed subtypes. Specifically, HPV was associated with a higher risk of squamous cell carcinoma, basal cell carcinoma, and melanoma. Notably, squamous cell carcinoma showed the highest HR among all cancer types studied. These findings highlighted the strong vital link between HPV and an increased risk of skin cancer, accentuating the need for targeted interventions and further research into preventive strategies.

Expanding on the discourse surrounding skin cancer risks, examining behavioral factors that potentially influence these outcomes becomes essential. A study revealed that a majority of men perceive tanning as beneficial to health and display a high level of affirmation towards it [26]. Moreover, reports on sun protection behavior suggest that, apart from wearing sunglasses, males often neglect sun protection measures and use sunscreen less frequently [27]. This contributes to a higher risk of skin cancer among men compared to women, consistent with our research findings [28]. Additionally, it is projected that skin cancer may become the leading cancer among men within the next 20 years. Furthermore, as age increases, the risk of skin cancer also escalates, particularly affecting the elderly population [29]. In our study, individuals aged 80 and above exhibited an HR as high as 58.20. This is likely attributed to the cumulative increase in ultraviolet radiation exposure with age, as well as exposure to ionizing radiation, immune suppression, and prolonged scar inflammation, among other contributing factors [30]. Research reports also indicated a significant aging trend in non-melanoma skin cancer (NMSC) compared to other types of cancers [31].

Urbanization and national development levels also impact the development of skin cancer. Although sunscreen is a part of many public health initiatives in high-income countries, its relatively high cost can be prohibitive [32]. Lack of financial capability often implies occupational exposure to outdoor work and sunlight entering homes with low-density materials, both of which may contribute to a higher incidence of skin cancer [33]. Ultraviolet (UV) light exposure has been confirmed to increase the risk of skin cancer. In Taiwan, generally, southern and Kaohsiung-Pingtung regions experience higher levels of sunlight and UVB exposure, followed by western areas [34]. Our research findings indicate a higher risk of skin cancer in the southern and Kaohsiung-Pingtung regions. A mouse experiment suggests a synergistic effect of HPV infection of the skin and UVB radiation in SCC formation [35]. HPV is believed to evade physiological DNA damage responses or delay DNA repair mechanisms [36], such as UV-induced cyclobutane pyrimidine dimer excision [37], ultimately favoring the accumulation and proliferation of cancer cells caused by UV damage [38].

Studies suggest that the risk of skin cancer may increase among agricultural and construction workers (ACW) due to their frequent exposure to high levels of solar ultraviolet radiation [8,39]. Moreover, over half of ACWs reported never using sunscreen when outdoors for more than 1 h on warm sunny days [40]. Similar trends were observed among fishermen, with a higher overall incidence of skin cancer associated with maritime occupations compared to the general population [41]. Exposure to carcinogenic chemicals such as asbestos and ultraviolet radiation at sea further elevated the risk of skin cancer, consistent with our research findings [42]. However, an interesting study revealed that individuals who worked for over 50 years, were exposed to sunlight for 21 to 28 h per week, and did not use any sun protection measures had lower incidences of vitamin D deficiency, which may exert a certain degree of inhibition against skin cancer [43]. Despite the potential protective effect of vitamin D against skin cancer, it is still recommended to implement sun protection measures to reduce the risk of skin cancer under typical circumstances [44].

A study conducted on a population-based cohort in the United States found a high correlation between hypertension and skin cancer [45]. However, other studies suggest that hypertension is not associated with skin cancer and may even reduce the risk of skin cancer [46]. This discrepancy could be attributed to the influence of hypertension medication on the risk of developing skin cancer. A study from Japan indicated that hydrochlorothiazide (HCT), a commonly used antihypertensive medication, increased the risk of non-melanoma skin cancer in elderly individuals with dark skin, highlighting the increased risk of SCC among HCT users in aging societies worldwide [47]. In Taiwan, common antihypertensive medications include angiotensin-converting enzyme inhibitors (ACEI), angiotensin II receptor blockers (ARB), beta-blockers (β-blocker), calcium channel blockers (CCB), and diuretics. Apart from the association of β-blockers with effective tumor immune response markers and their protective effects on human melanoma patients [46,47,48,49], other antihypertensive medications increase the risk of skin cancer [50,51,52,53]. Our results indicate an elevated risk of skin cancer with hypertension, but further research is needed to evaluate the associated risk of skin cancer after the use of antihypertensive medications in HPV patients.

Non-melanoma skin cancer (NMSC) is more prevalent compared to lung, breast, prostate, and colon cancers, with an annual incidence increase of 4–8% [54]. Among NMSC types, SCC is particularly common [55]. However, ultraviolet solar radiation is the primary risk factor for the development of skin SCC [56]. A study strongly suggested that environmental UV exposure, measured over the preceding 10 years, was associated with an increased prevalence of serum positivity for β-HPV [57]. Another mouse study indicates that papillomavirus (MmuPV1) mice, exposed to ultraviolet radiation (UVR), particularly the UVB spectrum, exhibit high susceptibility to MmuPV1-induced diseases. MmuPV1 mice develop warts when treated with UVB and further progress to SCC [58]. This further illustrates the complex relationship between HPV and UVB, where β-HPV types activate the transforming properties of E6 and E7 and suppress the tumor suppressor gene p53, leading to cancer development [59]. E7’s most famous target is the non-functional retinoblastoma protein (pRB), and direct binding of E7 to pRB impairs its cell regulatory function [60]. Moreover, high-dose UVB exposure leads to the loss of pRB, further contributing to cancer development [61]. Cells infected with HPV show continuous propagation of proliferative signals driven by the expression of E6 and E7. Aberrant growth stimulation mediated by E7-induced pRb breakdown can be stabilized through p53 [62], which is hindered by E6, with UVB-induced p53 mutations being a major risk factor [63]. Under UVB exposure, activation of multiple protein kinases, including ATR, ATM, CHK1, AMPK, p38K, and JNK, can decrease p53 activity by triggering the phosphorylation of certain residues [64]. A mouse study also confirms that targeting the IKKα/p53/PERP pathway may help prevent sunlight-induced skin photodamage [65]. Certain photosensitizing antihypertensive medications enhance skin responsiveness to sunlight, triggering delayed hypersensitivity reactions mediated by T cells, thereby increasing the risk of cutaneous SCC [66]. A study indicated a significantly higher intake rate of HCT or ACE inhibitors in the non-melanoma skin cancer patient population compared to conventional data DAK-G [51,67]. Thiazide diuretics may also increase the risk of skin cancer [68]. Our experiments suggest that HPV poses a higher risk for squamous cell carcinoma of skin cancer, possibly due to the influence of UVB exposure, resulting in the highest risk of cutaneous SCC.

Among the skin cancer subtypes, basal cell carcinoma (BCC) is the most common cancer in populations of European descent [69]. Exposure to substantial amounts of sunlight UVR, particularly UVB radiation, is a major risk factor for BCC [70]. Studies suggest a potential risk of γ-HPV in BCC patients, with the prevalence of γ-HPV being higher than α-, β-, and EV-HPV in BCC patients [71]. This demonstrates the impact of HPV on BCC. Furthermore, UVB exposure leads to melanoma, the deadliest form of skin cancer, with an estimated 60–70% of malignant melanoma cases attributed to exposure to ultraviolet radiation [72], and the incidence is on the rise, resulting in significant clinical and financial burdens [73]. HPV has been found in 58% of biopsy samples obtained from stage III and IV melanoma patients, indicating that HPV may serve as an indirect factor in melanoma development and regulate the transition of melanoma cells to a more aggressive phenotype [3,19]. A study from 2013 confirmed that although rare, the coexistence of malignant melanoma and facial BCC in the same specimen has been well documented [74]. Future research needs to further clarify the intertwined factors of HPV and UVB, as HPV is a high-risk factor for various types of skin cancer. While the association between high-risk HPVs and skin cancers is elusive, there is a possible interpretation for the results of our study: patients infected with mucosal types may have a higher chance of being affected by cutaneous types due to a dysregulated immune status.

In this study, the presence of HPV infection was defined using a range of ICD-9-CM codes, which encompass both low-risk and high-risk HPV types. This broad definition was employed due to the nature of the administrative data, which does not specify HPV subtypes in the diagnosed cases. While this method allows for a comprehensive inclusion of potential HPV-related cases, it is crucial to acknowledge that not distinguishing between oncogenic and non-oncogenic HPV types may have influenced the observed association between HPV and skin cancer risk. Several types of HPV have been well-documented for their roles in oncogenesis, particularly in anogenital cancers [14]; however, their roles in skin cancers are less clearly defined. By not differentiating the HPV types, our analysis potentially includes a wide spectrum of HPV-related skin conditions, which may or may not contribute to carcinogenesis. This broad inclusion is likely to affect the specificity of our findings but is justified given the exploratory nature of our study and the limitations of the available data. Future research with access to detailed genotyping of HPV could provide more definitive insights into which specific types are most significantly associated with skin cancer. This would allow for a more targeted approach in both research and clinical practice, potentially leading to better prevention and treatment strategies for HPV-related skin cancers.

Our inclusion of various comorbidities such as hypertension, diabetes mellitus, and chronic kidney disease in the regression analysis reflects an awareness of their potential influence on skin cancer risk. Recent studies have suggested that systemic conditions like hypertension may impact skin cancer development due to biological mechanisms such as inflammation or immunomodulation, which are influenced by these chronic diseases. For instance, hypertension has been associated with an increased risk of skin cancer, potentially linked to the photosensitizing effects of certain antihypertensive medications [46,47,48,49]. While direct skin-related conditions like actinic keratosis or immune diseases were not explicitly included as covariates, the selected comorbidities were chosen based on their documented implications in altering biological pathways that could affect skin health. This decision was also guided by the availability of robust, well-documented medical records within the National Health Insurance Research Database, ensuring the reliability of the data used for analysis.

Our study establishes a significant correlation between HPV and skin cancer risk, emphasizing a complex interplay of factors rather than a direct causative link. While HPV is known for its role in mucosal cancers, its association with skin cancer likely involves environmental and behavioral elements, notably UV exposure. The findings should not be interpreted as evidence that HPV vaccination would reduce skin cancer incidence, as the role of vaccines in preventing these cancers remains to be directly investigated. This study contributes to the foundational understanding of HPV’s potential role in skin carcinogenesis and emphasizes the necessity for further research to explore the mechanisms at play and the potential impact of preventive strategies.

One significant limitation of our study is the inability to establish a causal relationship due to the retrospective nature of the data. While we observed a strong association between HPV infection and increased risk of skin cancer, causality cannot be inferred. We controlled for several known risk factors, including age, gender, and occupational sun exposure, but acknowledge that other confounders, such as individual UV exposure and genetic predisposition, were not accounted for. This study still has some other limitations, including the lack of basic biochemical information about patients, basic lifestyle habits such as exercise or smoking and alcohol consumption, and information about medication usage. Nonetheless, through such matching and adjustments, we can still represent the patients’ living conditions and health status, leading to accurate results. Our strength lies in the data of HPV patients from across Taiwan concerning the correlation with skin cancer.

## 5. Conclusions

We conclude that HPV is highly associated with skin cancer risk, highlighting the importance of sun protection, especially in high-exposure areas such as southern Taiwan and the Kaohsiung-Pingtung region, as well as in high-exposure occupations like agriculture, forestry, and fishery. Additionally, further analysis of hypertension medications, particularly photosensitizing agents, is warranted to understand their impact on the susceptibility of HPV patients to skin cancer. While our findings suggest a significant association between HPV infection and skin cancer risk, these results should be interpreted as preliminary. Further research is necessary to establish causality and investigate the molecular mechanisms underlying this association.

## Figures and Tables

**Figure 1 cancers-16-02284-f001:**
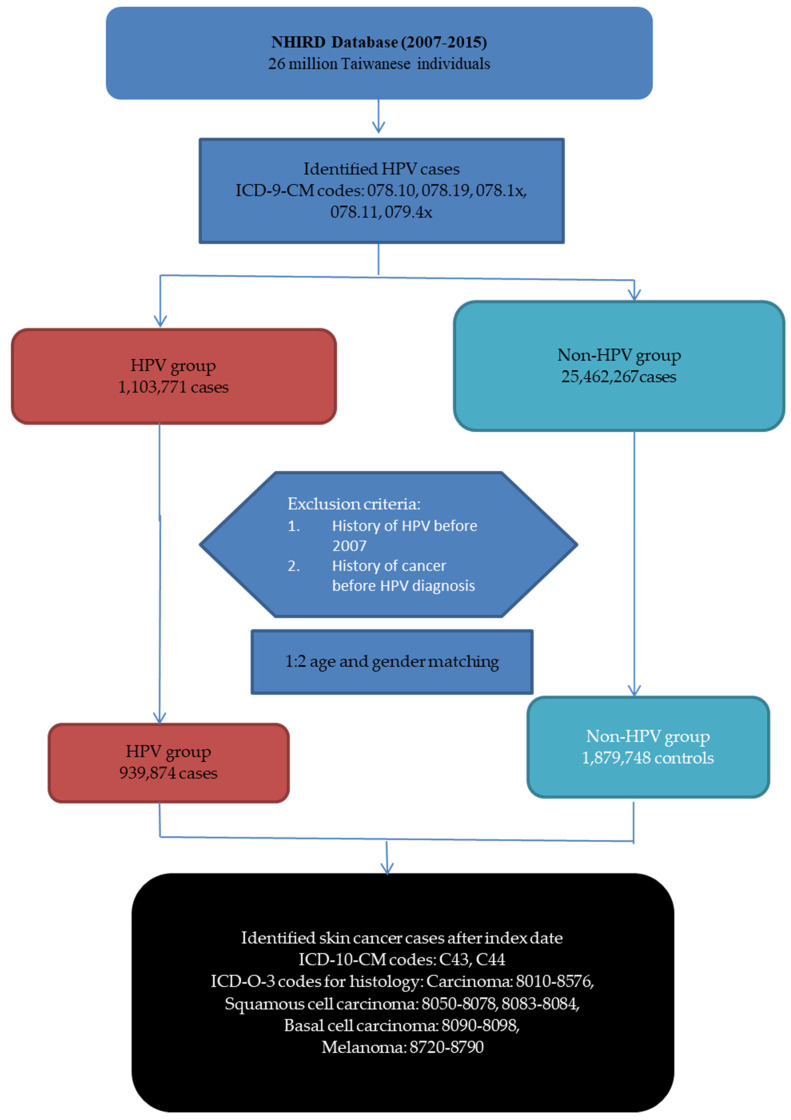
Study flow chart and participant disposition.

**Figure 2 cancers-16-02284-f002:**
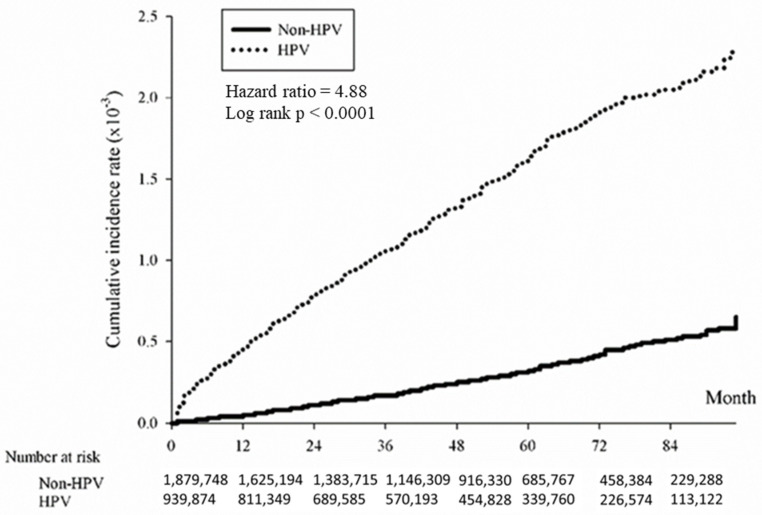
Kaplan-Meier method illustrating cumulative incidence of skin cancer in HPV-positive and HPV-negative patients.

**Figure 3 cancers-16-02284-f003:**
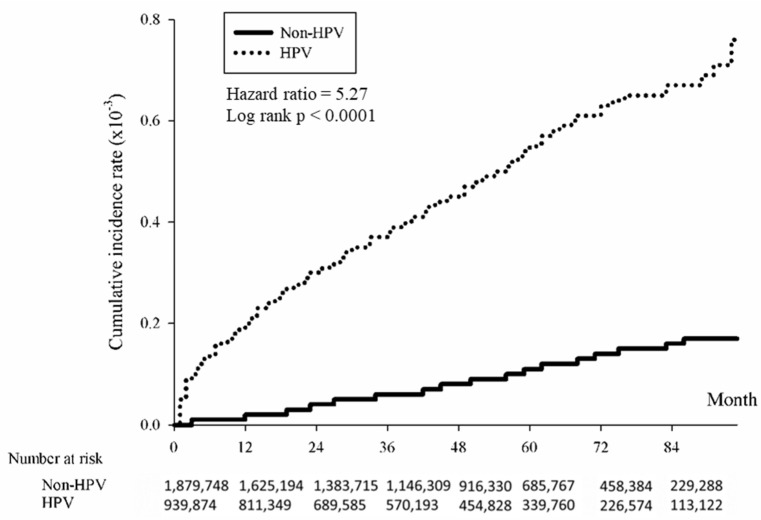
Kaplan-Meier method illustrating cumulative incidence of squamous cell carcinoma in HPV-positive and HPV-negative patients.

**Figure 4 cancers-16-02284-f004:**
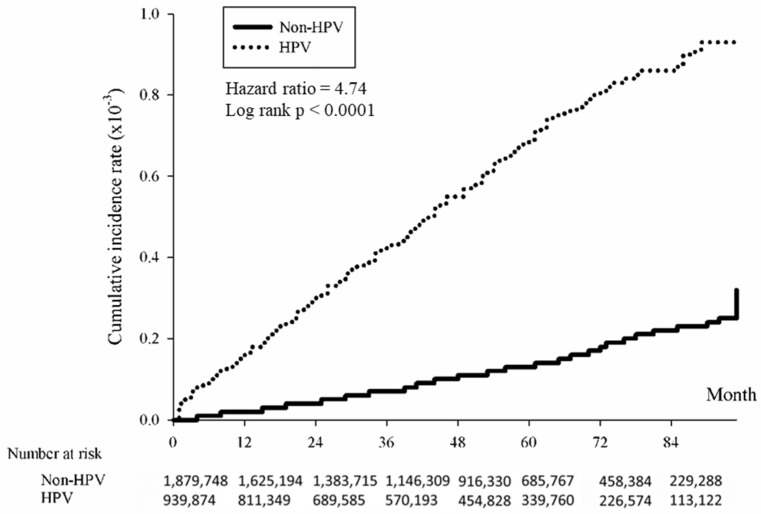
Kaplan-Meier method illustrating cumulative incidence of basal cell carcinoma in HPV-positive and HPV-negative patients.

**Figure 5 cancers-16-02284-f005:**
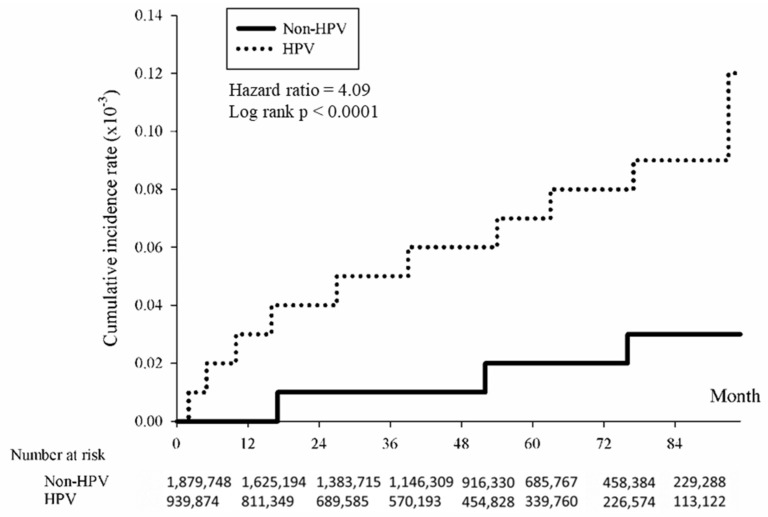
Kaplan-Meier method illustrating cumulative incidence of melanoma in HPV-positive and HPV-negative patients.

**Table 1 cancers-16-02284-t001:** Baseline characteristics of HPV and non-HPV groups.

	Non-HPV	HPV	*p*
Sex			
Male	934,908 (49.74%)	467,454 (49.74%)	1.0000
Female	944,840 (50.26%)	472,420 (50.26%)	
Age			
<20	513,000 (27.29%)	256,500 (27.29%)	1.0000
20–40	669,372 (35.61%)	334,686 (35.61%)	
40–60	469,150 (24.96%)	234,575 (24.96%)	
60–80	191,510 (10.19%)	95,755 (10.19%)	
>80	36,716 (1.95%)	18,358 (1.95%)	
Urbanization			
1 (Highest level)	542,433 (28.86%)	319,694 (34.01%)	<0.0001
2	587,017 (31.23%)	290,511 (30.91%)	
3	340,505 (18.11%)	162,859 (17.33%)	
4	246,844 (13.13%)	109,824(11.68%)	
5	36,687 (1.95%)	13,197 (1.40%)	
6	70,850 (3.77%)	23,620 (2.51%)	
7 (Lowest level)	55,412 (2.95%)	20,169 (2.15%)	
Area			
Taipei	671,431 (35.72%)	395,713 (42.10%)	<0.0001
Northern	279,894 (14.89%)	124,964 (13.30%)	
Central	347,664 (18.50%)	180,479 (19.20%)	
Southern	261,434 (13.91%)	107,447 (11.43%)	
Kaohsiung-Pingtung	276,130 (14.69%)	115,919 (12.33%)	
Eastern	43,195 (2.30%)	15,352 (1.63%)	
Insurance coverage			
Public insurance	111,753 (5.95%)	79,002 (8.41%)	<0.0001
Labor insurance	1183,084 (62.94%)	618,860 (65.84%)	
Agricultural, irrigation, Fishery associations	237,256 (12.62%)	94,224 (10.03%)	
Low-income	20,772 (1.11%)	7928 (0.84%)	
Unemployment	294,431 (15.66%)	122,182 (13.00%)	
Other	32,452 (1.73%)	17,678 (1.88%)	
Co-morbidity			
Ischemic heart disease	63,872 (3.40%)	39,334 (4.19%)	<0.0001
Hypertension	205,015 (10.91%)	112,709 (11.99%)	<0.0001
Stroke	40,087 (2.13%)	20,246 (2.15%)	0.2386
Diabetes mellitus	96,584 (5.14%)	46,782 (4.98%)	<0.0001
Abnormal liver function	91,126 (4.85%)	45,661 (4.86%)	0.7009
Renal failure	15,200 (0.81%)	8191 (0.87%)	<0.0001
GI bleeding	10,826 (0.58%)	5723 (0.61%)	0.0006
Hyperlipidemia	133,790 (7.12%)	83,114 (8.84%)	<0.0001
Chronic kidney diseases	25,740 (1.37%)	14,747 (1.57%)	<0.0001
COPD	32,326 (1.72%)	18,607 (1.98%)	<0.0001
Peptic ulcer	122,250 (6.50%)	74,302 (7.91%)	<0.0001
Gout	49,184 (2.62%)	28,685 (3.05%)	<0.0001

HPV, Human Papillomavirus; COPD, Chronic Obstruction Pulmonary Disease; GI, Gastrointestinal.

**Table 2 cancers-16-02284-t002:** Multiple Cox regression models of HPV for skin cancer.

	Adjusted HR	95% CI	*p*
HPV	5.22	4.70–5.80	<0.0001
Sex			
Male	1.20	1.10–1.34	<0.0001
Female	Reference		
Age			
<20	0.14	0.08–0.25	<0.0001
20–40	Reference		
40–60	4.15	3.36–5.15	<0.0001
60–80	19.55	15.84–24.13	<0.0001
>80	58.20	46.21–73.30	<0.0001
Urbanization			
1 (Highest level)	Reference		
2	0.95	0.82–1.09	0.4436
3	1.04	0.88–1.23	0.6471
4	0.99	0.83–1.18	0.8945
5	1.05	0.78–1.40	0.7656
6	0.90	0.70–1.17	0.4383
7 (Lowest level)	0.74	0.54–1.00	0.0518
Area			
Taipei	Reference		
Northern	1.17	0.97–1.41	0.0961
Central	1.31	1.12–1.53	0.0009
Southern	1.85	1.57–2.18	<0.0001
Kaohsiung-Pingtung	1.74	1.49–2.02	<0.0001
Eastern	1.58	1.14–2.18	0.0055
Insurance coverage			
Public insurance	1.12	0.92–1.36	0.2529
Labor insurance	Reference		
Agricultural, Irrigation, Fishery Associations	1.68	1.45–1.95	<0.0001
Low-income	1.08	0.58–2.03	0.8045
Unemployment	0.94	0.82–1.08	0.3778
Other	1.22	0.82–1.81	0.3261
Co-morbidity			
Ischemic heart disease	1.02	0.90–1.16	0.7600
Hypertension	1.26	1.12–1.42	<0.0001
Stroke	1.11	0.96–1.29	0.1658
Diabetes mellitus	1.14	1.00–1.30	0.0532
Abnormal liver function	0.97	0.83–1.14	0.7188
Renal failure	1.43	1.00–2.04	0.0531
Gastrointestinal bleed	1.14	0.82–1.59	0.4267
Hyperlipidemia	0.91	0.80–1.03	0.1198
Chronic kidney diseases	1.08	0.79–1.46	0.6401
COPD	1.16	0.99–1.36	0.0705
Peptic ulcer	1.06	0.94–1.20	0.3627
Gout	1.16	0.99–1.37	0.0644

Adjustment for sex, age, urbanization, area, insurance coverage, and co-morbidity. HPV, Human Papillomavirus; COPD, Chronic Obstruction Pulmonary Disease.

**Table 3 cancers-16-02284-t003:** HPV in multiple Cox regression models for subtypes of skin cancer.

	Non-HPV		HPV			
	Event	Incidence Rate ^†^	Event	Incidence Rate ^†^	Crude HR	Adjusted HR
Skin Cancer	501	0.56 (0.51–0.61)	1217	2.74 (2.59–2.90)	4.88 (4.40–5.42)	5.22 (4.70–5.80)
Carcinoma	423	0.47 (0.43–0.52)	1061	2.39 (2.25–2.54)	5.04 (4.50–5.64)	5.40 (4.82–6.06)
Squamous cell carcinoma	159	0.18 (0.15–0.21)	417	0.94 (0.85–1.03)	5.27 (4.39–6.32)	5.97 (4.96–7.19)
Basal cell carcinoma	212	0.24 (0.21–0.27)	500	1.13 (1.03–1.23)	4.74 (4.04–5.57)	4.88 (4.14–5.74)
Other carcinoma	52	0.06 (0.04–0.08)	144	0.32 (0.28–0.38)	5.57 (4.05–7.64)	5.96 (4.32–8.22)
Melanoma	28	0.03 (0.02–0.05)	57	0.13 (0.10–0.17)	4.09 (2.60–6.43)	4.36 (2.76–6.89)

Adjustment for sex, age, urbanization, area, insurance coverage, co-morbidity. HPV, Human Papillomavirus; † crude incidence rate, per 100,000 person-years.

## Data Availability

The data used in this research are not publicly available due to privacy and confidentiality restrictions. These data, which include patient health records and sensitive information, were accessed and analyzed under strict ethical and legal regulations. However, qualified researchers interested in accessing the data for replication or further investigation may submit requests to the National Health Insurance Administration, Ministry of Health and Welfare, Taiwan, following the established data access procedures.

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
