# Peer review of "Synergistic Effect of Human Papillomavirus and Environmental Factors on Skin Squamous Cell Carcinoma, Basal Cell Carcinoma, and Melanoma: Insights from a Taiwanese Cohort"

_cancers, 2024, doi:10.3390/cancers16122284_

Round 1

Reviewer 1 Report

Comments and Suggestions for Authors

The authors of this study look at the relationship between HPV and types of skin cancers using data from the national health insurance research database in Taiwan. They found significant associations between HPV and the development of cutaneous squamous cell carcinoma, basal cell carcinoma, and melanoma.  I had several questions to be clarified.

-          It is unclear what the defined the presence of HPV infection in these patients – more clarification than just a few ICD codes is needed. Are these oncogenic subtypes?

-          Not sure why they excluded individuals with HPV prior to 2007, and those who had cancer before the diagnosis of HPV.

-          Could the authors explain the medical comorbidities utilized for the regression analysis? These are various medical conditions, but it’s not clear how those relate at all. What about skin conditions such as actinic keratosis, immune disease, etc?

-          It seems that factors such as their job (ie if they worked in agriculture, construction, etc.) that might correlate with sun exposure should be accounted for in the matching.

-          Line 237 – those confidence intervals seem off. Please confirm.

-          Ultimately, it’s very hard to contextualize the impact of this study. The evidence that HPV is directly associated with skin cancer is not so direct, and it seems that there may be behaviors or exposures that are more relevant. For example, do the authors believe that use of the HPV vaccine will lead to a reduction of skin cancer? It seems that most of the associations are likely to have some other yet unknown effect rather than being direct. Thus, the impact of this manuscript is greatly dampened as it is correlative rather than causative.

Author Response

Comments and Suggestions for Authors

Reviewer 1

The authors of this study look at the relationship between HPV and types of skin cancers using data from the national health insurance research database in Taiwan. They found significant associations between HPV and the development of cutaneous squamous cell carcinoma, basal cell carcinoma, and melanoma.  I had several questions to be clarified.

Comment 1: It is unclear what the defined the presence of HPV infection in these patients – more clarification than just a few ICD codes is needed. Are these oncogenic subtypes?

Response 1:

We thank the reviewer for this insightful question. In response to inquiries regarding the specifics of HPV infection definitions in our study, it is crucial to clarify that HPV presence was identified through a series of ICD-9-CM codes. These include 078.10, 078.19, 078.1x, 078.11, and 079.4x. These codes represent a range of HPV infections, not limited to oncogenic types. The data used encompassed both clinical diagnoses of genital warts and other HPV-related conditions recorded in the National Health Insurance Research Database between 2007 and 2015.

  • We have added the descriptions of various ICD codes in the Materials and Methods section, under Subsection 2.2. Study Population:

HPV diagnosis was defined by ICD-9-CM codes: 078.10 (viral warts, unspecified), 078.19 (other specified viral warts), 078.1x (various forms of viral warts caused by HPV), 078.11 (condyloma acuminatum/genital warts), and 079.4x (unspecified viral and chlamydial infections)

  • Furthermore, we have amended the Discussion section to include:

In this study, the presence of HPV infection was defined using a range of ICD-9-CM codes, which encompass both low-risk and high-risk HPV types. This broad definition was employed due to the nature of the administrative data, which does not specify HPV subtypes in the diagnosed cases. While this method allows for a comprehensive inclusion of potential HPV-related cases, it is crucial to acknowledge that not distinguishing be-tween oncogenic and non-oncogenic HPV types may have influenced the observed association between HPV and skin cancer risk. Several types of HPV have been well-documented for their roles in oncogenesis, particularly in anogenital cancers; however, their roles in skin cancers are less clearly defined. By not differentiating the HPV types, our analysis potentially includes a wide spectrum of HPV-related skin conditions, which may or may not contribute to carcinogenesis. This broad inclusion is likely to affect the specificity of our findings but is justified given the exploratory nature of our study and the limitations of the available data. Future research with access to detailed genotyping of HPV could provide more definitive insights into which specific types are most significantly associated with skin cancer. This would allow for a more targeted approach in both research and clinical practice, potentially leading to better prevention and treatment strategies for HPV-related skin cancers.

Comment 2: Not sure why they excluded individuals with HPV prior to 2007, and those who had cancer before the diagnosis of HPV.

Response 2:

To address queries regarding the exclusion criteria applied in our study, particularly the exclusion of individuals with an HPV diagnosis prior to 2007 and those who had been diagnosed with cancer before HPV, we now include a detailed explanation in the "Materials and Methods" section of the revised manuscript. This explanation will clarify the rationale behind these decisions and how they contribute to the study's integrity and the validity of its findings.

  • The exclusion of individuals with a recorded diagnosis of HPV prior to 2007 was strategic, allowing us to clearly define the cohort's exposure period starting from 2007, thereby establishing a baseline from which to monitor new HPV diagnoses and subsequent skin cancer development. By setting 2007 as the exposure inception point, we aimed to differentiate between those exposed to HPV from those unexposed within the defined timeframe, ensuring a clear temporal relationship between HPV exposure and the development of skin cancer outcomes. Similarly, the exclusion of individuals with any cancer diagnosis prior to their HPV diagnosis was critical to eliminate confounding effects that pre-existing cancer might have on the study outcomes. This approach ensures that the observed associations between HPV and skin cancer are not influenced by previous cancers, which could skew the results and compromise the study's objectives to assess the direct impact of HPV on skin cancer risk.

Comment 3: Could the authors explain the medical comorbidities utilized for the regression analysis? These are various medical conditions, but it’s not clear how those relate at all. What about skin conditions such as actinic keratosis, immune disease, etc?

Response 3:

To address the inquiry raised, we have added in the Discussion:

  • Our inclusion of various comorbidities such as hypertension, diabetes mellitus, and chronic kidney disease in the regression analysis reflects an awareness of their potential influence on skin cancer risk. Recent studies have suggested that systemic conditions like hypertension may impact skin cancer development due to biological mechanisms such as inflammation or immunomodulation, which are influenced by these chronic diseases. For instance, hypertension has been associated with an increased risk of skin cancer, potentially linked to the photosensitizing effects of certain antihypertensive medications. While direct skin-related conditions like actinic keratosis or immune diseases were not explicitly included as covariates, the selected comorbidities were chosen based on their documented implications in altering biological pathways that could affect skin health. This decision was also guided by the availability of robust, well-documented medical records within the National Health Insurance Research Database, ensuring the reliability of the data used for analysis.

Comment 4: It seems that factors such as their job (ie if they worked in agriculture, construction, etc.) that might correlate with sun exposure should be accounted for in the matching.

Response 4:

In addressing the concern regarding the potential correlation between occupational exposure to sunlight and skin cancer risk, particularly for jobs that inherently involve significant outdoor activity such as those in agriculture or construction, our study has incorporated a methodical approach to account for these variations. This is addressed in our manuscript where we detail our matching criteria based on insurance coverage, which inherently correlates with specific types of employment.

We have amended the revised manuscript to explain this methodology by including in the "Materials and Methods" section under the subsection where the study population and matching criteria are described:

2.3. Accounting for Occupational Exposure in Matching Criteria

Recognizing the significance of occupational sun exposure as a potential confounding factor in the relationship between HPV and skin cancer, our study meticulously accounted for the participants' jobs by matching based on the type of insurance coverage. In Taiwan, insurance coverage is closely tied to one's occupation, with specific insurance categories for Labor, Agricultural, and Fishery workers, among others. This categorization allows for a control of occupational exposure, particularly to sunlight, which is more prevalent in certain job sectors. By utilizing this matching criterion, we ensured that the comparison between the HPV-positive group and the control group accounted for differences in occupational sun exposure risks. This approach enhances the reliability of our findings by ensuring that any observed differences in skin cancer incidence are more likely attributable to HPV status rather than confounding by occupational exposure.

Comment 5: Line 237 – those confidence intervals seem off. Please confirm.

Response 5: We would like to thank the reviewer for identifying these mistakes and have corrected the numbers accordingly and highlighted these revised confidence intervals in red.

Comment 6: Ultimately, it’s very hard to contextualize the impact of this study. The evidence that HPV is directly associated with skin cancer is not so direct, and it seems that there may be behaviors or exposures that are more relevant. For example, do the authors believe that use of the HPV vaccine will lead to a reduction of skin cancer? It seems that most of the associations are likely to have some other yet unknown effect rather than being direct. Thus, the impact of this manuscript is greatly dampened as it is correlative rather than causative.

Response 6: We thank the reviewer for pointing this out. We have added the valuable recommendation in our revised manuscript in the Discussion section:

Our study establishes a significant correlation between HPV and skin cancer risk, emphasizing a complex interplay of factors rather than a direct causative link. While HPV is known for its role in mucosal cancers, its association with skin cancer likely involves environmental and behavioral elements, notably UV exposure. The findings should not be interpreted as evidence that HPV vaccination would reduce skin cancer incidence, as the role of vaccines in preventing these cancers remains to be directly investigated. This study contributes to the foundational understanding of HPV's potential role in skin carcinogenesis and emphasizes the necessity for further research to explore the mechanisms at play and the potential impact of preventive strategies. 

Reviewer 2 Report

Comments and Suggestions for Authors

General comments

In this manuscript, with the aim of understanding the role of HPV as causal factor in skin cancer, the authors respectively analyzed the large history data retrieved from the National Health Insurance Research Database (NHIRD) regarding the malignant skin cancer, including squamous cell carcinoma, basal cell carcinoma and melanoma in Taiwan region. As the final conclusion, the authors drew that HPV is tightly associated with the risk of skin cancer, underscoring the importance of sun protection in particular for those living in high-exposure areas and occupations.

In actual fact, this study was retrospective cohort and the like in which I am not versed. I have been fascinated by basic research especially on biochemistry and basic pathology, instead of epidemiology. Given this, both the questions I raised and the perspective I took to question may be tangential or even far-fetched to this study. So here is the suggestion that I have to the authors when dealing with my questions, please take it or miss it at your discretion.

I have a vague recollection when taking the epidemiology course years ago that as for cohort designing, either retrospective or prospective, it needs exposed and un-exposed group. It seems that there was no mentioning at all that what the un-exposed group was in this manuscript. At least, I have not seen throughout. So, I am not sure whether it was OK. So, consultation with experts practicing epidemiological researches will be required and necessary.

Anyway, this paper is well-articulated and well prepared. Only minor questions I have for this paper from my angle were listed below, just for your reference

Minor questions

1.    Were there any inclusion and exclusion criteria you set up when enrolling the history information from the outset of this study? I have not found them after several rounds of perusals;

2.    Do you think will there be any necessary to provide the haematoxylin-eosin (so called HE) staining images representative of SCC, BCC and melanoma you included?

3.    Do you think will there be any necessary to provide the immunohistochemical staining of SCC, BCC and melanoma indicative of HPV infection with primary antibody targeting against E6/E7?

4.    Have you analyzed or specified the types, say high-risk or low-risk of HPV detected in skin cancer?

Author Response

Comments and Suggestions for Authors

Reviewer 2

General comments

In this manuscript, with the aim of understanding the role of HPV as causal factor in skin cancer, the authors respectively analyzed the large history data retrieved from the National Health Insurance Research Database (NHIRD) regarding the malignant skin cancer, including squamous cell carcinoma, basal cell carcinoma and melanoma in Taiwan region. As the final conclusion, the authors drew that HPV is tightly associated with the risk of skin cancer, underscoring the importance of sun protection in particular for those living in high-exposure areas and occupations.

In actual fact, this study was retrospective cohort and the like in which I am not versed. I have been fascinated by basic research especially on biochemistry and basic pathology, instead of epidemiology. Given this, both the questions I raised and the perspective I took to question may be tangential or even far-fetched to this study. So here is the suggestion that I have to the authors when dealing with my questions, please take it or miss it at your discretion.

Comment 1: I have a vague recollection when taking the epidemiology course years ago that as for cohort designing, either retrospective or prospective, it needs exposed and un-exposed group. It seems that there was no mentioning at all that what the un-exposed group was in this manuscript. At least, I have not seen throughout. So, I am not sure whether it was OK. So, consultation with experts practicing epidemiological researches will be required and necessary.

Response 1: We thank the reviewer for highlighting the need for a clearer distinction between exposed and unexposed groups in our cohort study design. In response, we have revised the manuscript to explicitly define the unexposed group within the "Materials and Methods" section:

After implementing a 1:2 age and gender matching, where each HPV-positive individual (exposed group) was compared to two controls without a recorded HPV diagnosis (unexposed group) from the same dataset, we included 939,874 cases in the HPV group and 1,879,748 controls in the non-HPV group.

Anyway, this paper is well-articulated and well prepared. Only minor questions I have for this paper from my angle were listed below, just for your reference

Minor questions

  1. Were there any inclusion and exclusion criteria you set up when enrolling the history information from the outset of this study? I have not found them after several rounds of perusals;

Response 1: For this study, inclusion criteria were set to ensure that all participants had a clear diagnosis of HPV between 2007 and 2015, as determined by specific ICD-9-CM codes (078.10, 078.19, 078.1x, 078.11, 079.4x). To focus on the new incidence of skin cancer post-HPV diagnosis, we excluded individuals with any cancer diagnosis prior to their first recorded HPV diagnosis to avoid confounding by pre-existing conditions. Additionally, individuals with a history of HPV prior to 2007 were excluded to establish a clear baseline for observing new cases, ensuring that the temporal relationship between HPV exposure and subsequent skin cancer development could be accurately assessed. For the amendments, please refer to the revised manuscript under Materials and Methods section 2.2 Study Population.

  1. Do you think will there be any necessary to provide the haematoxylin-eosin (so called HE) staining images representative of SCC, BCC and melanoma you included?

Response 2: Thank you for the suggestion to include haematoxylin-eosin (H&E) staining images of SCC, BCC, and melanoma. We acknowledge that such images can provide valuable morphological insights into the histological characteristics of these cancers. However, it is important to clarify that our study is purely epidemiological in nature, focusing on population-level data rather than individual tissue specimens. The research utilizes the National Health Insurance Research Database, which comprises coded health data and does not involve direct analysis of pathological specimens. Consequently, the inclusion of H&E staining images is not applicable in this context, as our analysis is based on ICD codes and statistical evaluations of recorded diagnoses, not on physical tissue examination. This methodological focus supports our study's goal of assessing long-term risk associations at the population level without the direct examination of biological specimens.

  1. Do you think will there be any necessary to provide the immunohistochemical staining of SCC, BCC and melanoma indicative of HPV infection with primary antibody targeting against E6/E7?

Response 3: As above.

  1. Have you analyzed or specified the types, say high-risk or low-risk of HPV detected in skin cancer?

Response 4: Thank you for inquiring about the specificity of HPV types—high-risk versus low-risk—in relation to skin cancer in our study. We recognize the importance of distinguishing between these subtypes due to their different oncogenic potentials. To address this, we have added a section to the "Discussion" part of our manuscript, where we elaborate on the types of HPV implicated in our findings:

In this study, the presence of HPV infection was defined using a range of ICD-9-CM codes, which encompass both low-risk and high-risk HPV types. This broad definition was employed due to the nature of the administrative data, which does not specify HPV subtypes in the diagnosed cases. While this method allows for a comprehensive inclusion of potential HPV-related cases, it is crucial to acknowledge that not distinguishing between oncogenic and non-oncogenic HPV types may have influenced the observed association between HPV and skin cancer risk. Several types of HPV have been well-documented for their roles in oncogenesis, particularly in anogenital cancers [14]; however, their roles in skin cancers are less clearly defined. By not differentiating the HPV types, our analysis potentially includes a wide spectrum of HPV-related skin conditions, which may or may not contribute to carcinogenesis. This broad inclusion is likely to affect the specificity of our findings but is justified given the exploratory nature of our study and the limitations of the available data. Future research with access to detailed genotyping of HPV could provide more definitive insights into which specific types are most significantly associated with skin cancer. This would allow for a more targeted approach in both research and clinical practice, potentially leading to better prevention and treatment strategies for HPV-related skin cancers.

Round 2

Reviewer 1 Report

Comments and Suggestions for Authors

I realize that the authors have done their best methodologically within the context of the database being queried. However, the fact remains that this is simply an association that could have actually very little to do with HPV infection as a causative source. Given the lack of data supporting the causal mechanism of carcinogenesis and the inability to control for truly known risk factors of cutaneous malignancy, it seems that this study is quite premature. 

Author Response

Author's Reply to Reviewer 1

Response to Reviewer:

We appreciate the reviewer’s concerns and agree that while our study demonstrates an association between HPV infection and skin cancer, establishing a direct causal relationship remains challenging due to the observational nature of the data. We have made the following amendments to the manuscript to address these limitations and provide a more balanced interpretation of our findings.

  1. In the Discussion, we expanded the limitations section:

"One significant limitation of our study is the inability to establish a causal relationship due to the retrospective nature of the data. While we observed a strong association between HPV infection and increased risk of skin cancer, causality cannot be inferred. We controlled for several known risk factors, including age, gender, and occupational sun exposure, but acknowledge that other confounders, such as individual UV exposure and genetic predisposition, were not accounted for. Future studies should aim to incorporate these variables and employ longitudinal designs to better establish causality."

  1. In Conclusion, we included a statement highlighting the preliminary nature of the findings and the need for further research:

"While our findings suggest a significant association between HPV infection and skin cancer risk, these results should be interpreted as preliminary. Further research is necessary to establish causality and investigate the molecular mechanisms underlying this association."

By addressing these points and making the corresponding amendments to our manuscript, we aim to provide a more comprehensive and balanced interpretation of our study’s findings and limitations. We hope these changes adequately address your concerns.

Sincerely,

Stella Chin-Shaw Tsai, MD, PhD, FACS